# Bacteriophages Roam the Wheat Phyllosphere

**DOI:** 10.3390/v14020244

**Published:** 2022-01-26

**Authors:** Laura Milena Forero-Junco, Katrine Wacenius Skov Alanin, Amaru Miranda Djurhuus, Witold Kot, Alex Gobbi, Lars Hestbjerg Hansen

**Affiliations:** 1Department of Plant and Environmental Sciences, University of Copenhagen, 1871 Frederiksberg, Denmark; ksa@plen.ku.dk (K.W.S.A.); amaru@plen.ku.dk (A.M.D.); wk@plen.ku.dk (W.K.); alex.gobbi@plen.ku.dk (A.G.); 2Department of Environmental Science, Aarhus University, 4000 Roskilde, Denmark

**Keywords:** viral metagenomics, bacteriophage, phyllosphere, *Triticum aestivum*

## Abstract

The phyllosphere microbiome plays an important role in plant fitness. Recently, bacteriophages have been shown to play a role in shaping the bacterial community composition of the phyllosphere. However, no studies on the diversity and abundance of phyllosphere bacteriophage communities have been carried out until now. In this study, we extracted, sequenced, and characterized the dsDNA and ssDNA viral community from a phyllosphere for the first time. We sampled leaves from winter wheat (*Triticum aestivum*), where we identified a total of 876 virus operational taxonomic units (vOTUs), mostly predicted to be bacteriophages with a lytic lifestyle. Remarkably, 848 of these vOTUs corresponded to new viral species, and we estimated a minimum of 2.0 × 10^6^ viral particles per leaf. These results suggest that the wheat phyllosphere harbors a large and active community of novel bacterial viruses. Phylloviruses have potential applications as biocontrol agents against phytopathogenic bacteria or as microbiome modulators to increase plant growth-promoting bacteria.

## 1. Introduction

The phyllosphere comprises all the above-ground sections of plants. It is regarded as a harsh environment for microbial life due to fluctuations in water, nutrient availability, and high UV exposure [1]. This habitat is colonized by complex microbial communities, including bacteria, fungi, and archaea [1], although microbial densities are generally lower than below-ground plant portions [2]. The most abundant members of this community are considered to be phyllosphere-associated bacteria with an estimate of 10^6^ to 10^7^ cells per square centimeter [3]. Phyllobacteria can provide beneficial effects for plant health, such as protection against pathogens, nitrogen fixation, and phytohormone biosynthesis [4]. Thus, by acquiring more knowledge on how phyllosphere bacterial communities are modulated, new applications can be proposed to tackle the current agricultural challenges [5,6,7]. Bacteriophages, as predators of bacteria, are significant players in shaping microbial communities [8,9,10,11,12]. Only one published study has looked at the impact of bacteriophages present in the tomato phyllosphere on the associated bacterial community composition and abundance [13]. This study demonstrates that the bacteriophage community can alter the relative abundance of the dominant bacterial taxa. Although previous studies have surveyed phyllosphere viral communities [14,15,16,17,18,19], their methodologies and analyses have focused on eukaryotic viruses, thus overlooking bacteriophages. Therefore, knowledge about the bacteriophage community regarding species diversity and genomic potential does not exist in the literature.

In graminaceous plants, the flag leaf is the last leaf to emerge, and it is located below the spike. Likewise, the penultimate leaf is the second last leaf to emerge, situated below the flag leaf [20]. On wheat, the flag leaf dramatically influences the crop yield, as it supplies 30–50% of the energy to fuel grain filling [21]. Therefore, we sampled the flag and penultimate leaves of the winter wheat plant, *Triticum aestivum*, and extracted the associated viral and microbial fractions. We generated double-stranded DNA (dsDNA) and single-stranded DNA (ssDNA) viral metagenomes to characterize a phyllosphere bacteriophage community for the first time. In addition, the microbial fraction was sequenced to profile the microbial community and link the phylloviruses to potential bacterial hosts.

## 2. Materials and Methods

### 2.1. Sample Collection and Processing

A total of 500 flag leaves (FL) and 500 penultimate leaves (PL) from the wheat plant cultivar Elixir was randomly collected on 8 June 2020, when the plants were in the ripening/maturity stage [22]. Sampling was performed at plot 41-0 (55.6839, 12.2903) from the organic research fields of the University of Copenhagen, near Vridsloesemagle, Denmark. Leaves were cut up and shaken at 200 rpm in 1 L SM-buffer (100 mM NaCl, 8 mM MgSO_4_, 50 mM Tris-Cl) for 1 h at 4 °C to transfer the bacteriophages from the leaves into the buffer for viral-like particle (VLP) concentration and DNA extraction. After manually removing the leaf cuts, the SM-buffer was centrifuged for 10 min at 10,000× *g* followed by filtering using 5.0 μm syringe filters (PVDF, Merck Millipore, Darmstadt, Germany) to remove plant cells, chloroplasts, and large microbes. A total of 50 mL of this filtrate was saved as an input for the DNA extraction of the microbial fraction. The remaining filtrate was further filtered using 0.45 μm syringe filters to obtain the viral fraction. 

### 2.2. DNA Extraction and Amplification

Polyethylene glycol (PEG) precipitation was carried out to concentrate a 1 L sample down to a 10 mL SM-buffer containing the viral fraction, as explained in the protocol ‘Harvesting, Concentration, and Dialysis of Phage’ from The Actinobacteriophage database (https://phagesdb.org/media/workflow/protocols/pdfs/Harvesting_Concentration_and_Dialysis_of_Phage_PDF.pdf, last accessed 4 January 2022). DNA extraction was carried out as detailed in [23]. Briefly, the two 10 mL concentrated virome samples (FL and PL) were treated with 83 U of DNase I (A&A Biotechnology, Gdynia, Poland) for 1 h at 37 °C with CaCl_2_ + MgCl_2_ added in a 10 mM concentration to remove free DNA. This was followed with 18 U of proteinase K treatment (A&A Biotechnology, Gdynia, Poland) and 0.1% (*w*/*v*) sodium dodecyl sulfate (SDS) for 1 h at 55 °C to remove phage capsids and other proteins. The treatment was followed by Clean and Concentrator-5 (Zymo Research, Irvine, CA, USA) DNA extraction (FL and PL samples). Then 10 ng of viral DNA was MDA amplified (FL_mda_ and PL_mda_ samples) using a Cytiva illustra^TM^ Ready-To-Go^TM^ GenomiPhi V3 DNA Amplification Kit (Cytiva, Marlborough, MA, USA). Finally, the microbial fraction was centrifuged for 10 min at 10,000× *g*. The pellet was re-suspended in the lysis buffer from the DNeasy Powerlyzer Powersoil Kit (QIAGEN, Hilden, Germany), and the microbial DNA was extracted according to the manufacturer’s instructions (samples FL_mf_ and PL_mf_).

### 2.3. Library Preparation for Viral and Bacterial Samples for Sequencing

The concentration and purity of the DNA from both the viral and microbial fractions were measured using the Qubit 2.0 fluorometer (Life Technologies, Carlsbad, CA, USA) and NanoDrop 2000 spectrophotometer (Thermo Scientific, Waltham, MA, USA), respectively. Libraries for the native (FL, PL) and the MDA amplified (FL_mda_, PL_mda_) viromes, as well as the microbial fraction (FL_mf_ and PL_mf_), were prepared according to the manufacturer’s instructions using the Nextera^®^ XT DNA kit (Illumina, San Diego, CA, USA). The six libraries were sequenced as paired-end 2 × 150 bp reads on an Illumina NextSeq500 platform using the Mid Output Kit v2 (300 cycles).

### 2.4. Quality Control and Assembly

Raw sequencing quality was assessed using FastQC v.0.11.9 [24]. A Trimmomatic v.0.39 [25] was used for the removal of low-quality bases and adapters. Additionally, reads mapping to ΦX174 were removed using BBDuk from BBtools suite v.38.86 [26]. K-mer rarefaction curves were created using bbcountunique (BBtools) to assess the sufficiency of the sequencing effort. Quality controlled reads were error corrected and digitally normalized to a target coverage of 100× using BBnorm (BBtools suite). Digital normalization was performed to remove redundant reads, thus reducing memory requirements on the assembly step [27,28]. The k-mer abundances on the sequencing reads were calculated, and reads with an estimated mean coverage over a user-defined threshold were rejected. Assembly was performed using metaSPAdes v.3.14.0 [29] without the error correction step. The resulting contigs were filtered to a minimum contig size of 1000 bp and 2× coverage.

### 2.5. Viral Identification and Species Level Clustering

VirSorter v.2.0 [30] was run on the assembled contigs from the viral samples (FL, PL, FL_mda_, and PL_mda_) to identify viral contigs and classify them into dsDNA or ssDNA phages. The viral contigs identified were then dereplicated by clustering into virus operational taxonomic units (vOTUs) using the stampede-clustergenomes script (https://bitbucket.org/MAVERICLab/stampede-clustergenomes/, last accessed 11 March 2019). They were clustered at approximately species level following the MIUViG guidelines (≥95% nucleotide identity across ≥85% of their lengths) [31], and the longest member of each cluster was selected as the representative contig. The completeness of the cluster representatives was predicted using CheckV v0.6.0 [32], and vOTUs were defined as representative contigs with a length > 10 kb or labeled as complete/high-quality viral genomes [33]. Identification of lysogenic vOTUs was performed on complete/high-quality viral genomes using VIBRANT v1.2.1 [34].

### 2.6. Microbial Fraction Profiling and Host Assignment

The relative abundance of bacterial communities was performed by classifying FL_mf,_ and PL_mf_ reads at the family level using Kraken v2.1.1 [35] with the minikraken2_v28GB_201904 database. Visualization of relative abundances was done using Pavian v1.0 [36]. Two sets of CRISPR spacers were used to link viral contigs with potential bacterial host families. The first set was derived from CRISPR spacers identified from the assembled contigs of the microbial fraction samples (FL_mf_ and PL_mf_), using MinCED (https://github.com/ctSkennerton/minced/, last accessed 22 September 2020). The taxonomy of the FL_mf_ and PL_mf_ contigs was assigned, at the family level, using the MMSeqs2 v12.113e3 [37] taxonomy module with a minimum query coverage of 0.3. The second CRISPR-spacer set was the Dion database [38] containing over 11 million spacers from reference bacterial genomes. The bacterial taxonomic lineages from the Dion set were recovered using ETE v3.1.2 [39]. Phage-host relationships were identified by matching vOTU representative contigs to CRISPR spacers using SpacePHARER release 5 [40]. At least 80% of the CRISPR hits matching a vOTU had to belong to the same bacterial family. 

### 2.7. vOTU Taxonomy Assignment

First, vOTUs were aligned using nucleotide BLAST + 2.9.0 [41] to the viral RefSeq v94 [42] database and the IMG/VR 3.0 database [43]. Database sequences with alignments covering >40% of a vOTU and a subject/query length ratio of >0.8 were used as references to perform taxonomy assignments at the species and genus levels (referred to as the viral reference set). At the species level, viral taxonomy was assigned by clustering the vOTUs and the viral reference set at 85% coverage and 95% identity, using the same method described in the above section. Furthermore, vOTUs were grouped into genus-level viral clusters (VC) using vConTACT2 v0.9.19 [44] via protein profile clustering using viral RefSeq v94 sequences and the viral reference set. 

### 2.8. Relative Abundances and Average Read Depths of the Viral Populations

Quality-controlled reads were mapped against the vOTUs using bbmap v.2.3.5 [45]. bbmap was set with a minimum identity cutoff of 90%, and ambiguous mapping reads were assigned to the first encountered best site. SAMtools v.1.10 [46] was used to count the number of mapped reads per vOTU, then divided by contig length and normalized by sequencing depth (million reads). Contigs were required to have more than 75% breadth coverage, meaning that a vOTU had to be covered on more than 75% of its length to be considered present in a sample [33]. Contig breadth coverage was calculated using genomecov from BEDtools v.2.29.2.0 [47]. The fraction of mapped reads was determined by calculating the percentage of quality-controlled reads mapped to assembled contigs, viral contigs, and vOTUs. Coverage plots from BAM files were created using weeSAM version 1.5 (https://github.com/centre-for-virus-research/weeSAM, last accessed 22 September 2020).

### 2.9. Viral Particle Estimation

The following formula was used to estimate the number of viral particles per leaf:# viral particlesleaf=Xng×F×6.0221×1023 moleculesmole(NKbp×660 gmole × bp)×109 ngg
where *X* corresponds to the amount of DNA extracted from the viral fraction (0.13 ng/leaf), *F* corresponds to the fraction of reads that map to viral contigs (0.84), and *N* corresponds to 49.35 kbp, which was the calculated average length of the complete viral genomes recovered in this study (weighted by their relative abundance).

### 2.10. Manual Recovery of Two Complete Viral Genomes

Viral metagenomes consisting of abundant and closely related viral strains are challenging to assemble using short-reads [28,33]. The presence of regions of high diversity, such as genomic islands, results in unsuccessful branch resolutions in assembly graphs, and thus, fragmented assemblies. We recovered the high-quality genomes of two top rank-abundant and microdiverse viral populations by assembling subsets of each sample, thus reducing the graph complexity. The subsets of each sample were made with 10, 20, 30, 40, 50, 60, 70, 80, and 90% of the reads using the script reformat.sh from the BBtools suite. The subsampled reads were normalized and assembled as described above. Two complete viral genomes, namely NODE_40, recovered from the 80% subsampling of the PL, and APSE_wheat, from the 10% subsampling of PL_mda_, were included in the viral population clustering step.

### 2.11. Comparative Genomics of APSE Variants

The wheat APSE genome was compared against 15 other previously described APSE genomes [48]. The detection of open reading frames and annotation of viral contigs were carried out using VIGA a0c3ca7 (https://github.com/EGTortuero/viga/, last accessed 23 March 2020). Finally, the gene cluster comparison of the APSE subtypes was calculated and visualized using clinker v0.0.20 [49]. 

## 3. Results

### 3.1. Diverse Phyllosphere Associated Bacteriophage Communities Can Be Successfully Recovered Using Viral Metagenomics

Four whole-genome shotgun viral metagenomes were generated. Firstly, the DNA from the viral fraction of flag and penultimate leaves was extracted and directly sequenced (samples FL and PL, respectively). Furthermore, as ssDNA phages are selectively amplified by multiple displacement amplification (MDA) [50], two MDA amplified viral metagenomes were also sequenced (samples FL_mda_ and PL_mda_).

The viral fraction extracted from the FL and PL samples resulted in 91.6 ng and 37.2 ng of DNA, respectively (Appendix A). The apparent saturation of the k-mer-based rarefaction curves suggested that the sequencing effort was sufficient to recover the most abundant phylloviruses on the four viral fraction samples (FL, PL, FL_mda_, and PL_mda_; Appendix A). The metagenomic assemblies of these viromes produced a total of 25,977 contigs. A total of 87.2% of the quality-controlled reads can be mapped back to such contigs. The remaining 12.8% represents singletons, meaning non-overlapping reads that could not be used during assembly. Additionally, they could represent reads that were assembled, but they produced short or low coverage contigs, and thus were not able to fulfill the filtering threshold. 

In total, 15,739 contigs (60.6%) were identified as viral based on the detection of genomic features and unique viral markers [51]. Such viral contigs had a cumulative length of 72.9 million bp (Mbp). No less than 84.2% of the quality-controlled reads mapped to these viral contigs, suggesting that the viral enrichment method was successful, and most of the extracted DNA was of viral origin. The remaining 10,238 contigs (39.4%) corresponded to 3% of the fraction of mapped reads. These could represent assembly artifacts, host/microbial DNA remnants, or, likely, short viral fragments with no homologs in viral reference databases. 

Furthermore, the viral contigs were clustered into 876 species-rank virus groups referred to as viral operational taxonomical units (vOTUs) [31]. The vOTU clustering was made following current guidelines of ≥95% average nucleotide identity and ≥85% sequence coverage [31]. Moreover, they only included viral contigs of length ≥10 kbp [33] or viral contigs predicted as high-quality/complete viral genomes. Notably, 72.2% of the quality-controlled reads mapped to the 876 vOTUs, even though they constituted only 3.4% of the number of assembled contigs (Figure 1A). This suggests that most of the genomic diversity of the viral fraction is represented by these 876 vOTUs.

A total of 65 vOTUs was annotated as complete phage genomes, 107 as high quality, and 139 as medium quality (Figure 1B, Appendix A). The length of the complete viral genomes ranged from ssDNA phages of 2135 bp to jumbo phages of 354,470 bp (Appendix A). Out of the 172 predicted high-quality and complete vOTUs, 41 (23.8%) were identified as having a lysogenic, and 131 (76.2%) as having a lytic lifestyle. After mapping the clean reads to the vOTUs, 867 could be identified in at least one sample (Figure 1C, Appendix A). The remaining nine vOTUs did not pass the required breadth coverage thresholds to label them as being present. We identified 400 vOTUs in the FL and 698 vOTUs in the PL samples. In contrast, we found 183 and 368 vOTUs in the FL_mda_ and PL_mda_, respectively, showing an apparent loss of viral diversity consistent with known MDA amplification biases [52] (Figure 1C). A total of 46 vOTUs was unique to the amplified fractions (FL_mda_ and PL_mda_) of which five were high-quality ssDNA viral genomes (Figure 1C). On the contrary, no high-quality ssDNA genomes were present on the FL and PL samples. There were 75 vOTUs shared between the four viromes (FL, PL, FL_mda_, and PL_mda_), and we observed a higher viral diversity in the penultimate leaf (PL, PL_mda_) compared to flag leaf (FL, FL_mda_) samples. Based on the amount of DNA extracted from the leaves (Appendix A) and the average length of the complete genomes recovered from this study, we estimated a minimum of 1.0 × 10^5^ viral particles per cm^2^ or, on average, 2.0 × 10^6^ viral particles per leaf. Although bacteriophages usually outnumber bacteria 10 to 1 [53], this estimation of viral particles per cm^2^ is 10- to 100-fold lower than that of the phyllobacteria, reported between 1.0 × 10^6^ to 1.0 × 10^7^ [3]. Nevertheless, this is consistent with estimations made on soil samples, where the proportion of bacteriophages is also 10- to 100-fold lower than that of bacteria [53]. 

### 3.2. The Wheat Phyllosphere Microbial Fraction Is Dominated by Members of the Pseudomonadaceae Family 

We were able to assign 69.9% of the microbial fraction reads (FL_mf_ and PL_mf_) into a taxonomic rank. The taxonomic profile indicates that the dominant phyla on the wheat phyllosphere samples are Proteobacteria, Bacteriodetes, Actinobacteria, and Firmicutes, in that order (Figure 2A). Furthermore, the most abundant members of this community belong to the Xanthomonadaceae and Pseudomonadaceae families as 33.2% and 25.1% of the reads are assigned to those taxa, respectively. (Figure 2A). Therefore, the dominant members of the sampled microbial community are consistent with previous studies of the wheat phyllosphere microbiome [54,55].

### 3.3. The Phyllosphere Associated Viral Community Is Predicted to Infect the Dominant Bacterial Families on the Microbial Fraction

We used CRISPR spacer-based in silico host predictions to link the vOTUs to their probable bacterial host family [38]. In this way, we detected nucleotide matches between two sets of predicted CRISPR spacers and the vOTUs. The metagenomic assembly of microbial fraction samples (FL_mf_ and PL_mf_) produced 105,692 contigs with a cumulative assembly length of 337.4 Mbp. From these microbial contigs, we predicted a set of 493 CRISPR spacers that matched 27 (3.1%) vOTUs. Of those, 17 were assigned to bacterial hosts from the Erwinaceae family. Furthermore, we matched 698 (79.0%) additional vOTUs to CRISPR spacers from the Dion database, consisting of over 11 million CRISPR spacers. Overall, we identified 78 (8.9%) vOTUs targeting Pseudomonadaceae and 21 (2.4%) targeting Xanthomonadaceae, the two most abundant bacterial families (Figure 2B). This suggests that the wheat bacteriophage community infects the dominant members of the bacterial community. Thus, it is a potential driver of the microbial composition, consistent with previous findings on the tomato phyllosphere [13]. Interestingly, 154 vOTUs (17.7%) were identified to infect the Enterobacteriaceae family, which represented as little as 0.9% of the microbial fraction’s relative abundance and was primarily found in the FL_mf_ (Figure 2B, Appendix A). This high number of vOTUs could be explained due to a CRISPR-spacer database bias, as 75.0% of the spacers on the Dion database belong to the Enterobacteriaceae family [38].

### 3.4. The Wheat Phyllosphere Harbors a Distinct Bacteriophage Community

The uniqueness of the vOTUs was investigated at the genus and species level by comparing the vOTUs against viral sequences on the IMG/VR 3.0 database. This database includes a collection of more than 2 million cultivated and uncultivated viral sequences, including viral genomes. At the species level, 28 (3.2%) vOTUs were clustered with reference viral sequences. Therefore, as much as 848 (96.8%) vOTUs correspond to novel viral species. Furthermore, vConTACT2 [44] was used to calculate a gene-sharing profile of the vOTUs, viral RefSeq genomes, and IMG/VR viral sequences (Figure 3A). This network illustrates the protein-profile relatedness, which places the wheat phyllosphere virome into a taxonomic context. These protein profiles are used by vConTACT2 to group viral sequences into approximate genus-level clusters. Only 29 (3.3%) vOTUs were assigned taxonomy at genus-level, as they clustered together in the same viral cluster (VC) as a viral RefSeq genome. The majority of the wheat vOTUs are connected to reference genomes belonging to the *Caudovirales* order, also known as tailed bacteriophages. Notably, we identified 17 discrete groups of interconnected nodes that include only wheat phyllosphere vOTUs (dashed-line circles, Figure 3A). These phyllosphere-only groups represent a total of 77 vOTUs. For instance, NODE_40, one of the most abundant vOTUs found in the PL sample (Figure 4A), is part of one of these phyllosphere-only groups (Figure 3B). This group includes 30 vOTUs (species-level taxonomy), which are clustered in 13 VCs (genus-level taxonomy). This suggests there are numerous and diverse viral species unique to the wheat phyllosphere, perhaps with hitherto unexplored genomic potential.

### 3.5. The Top Rank-Abundant Phages Are Distinct on Each Leaf Type

The relative abundance of bacteriophage communities usually follows a power-law distribution [56], where few viral strains are dominant at a given sampling point. Hence, we examined the dominant vOTUs in the FL and PL samples, defined as the ten-top rank-abundant vOTUs. A total of 19 dominant vOTUs was explored as one vOTU, namely APSE_wheat, was top rank-abundant in both leaf types. This indicates that the top rank-abundant phyllosphere vOTUs are different across leaf types. Only three dominant vOTUs could be classified to the species level and six to the family level (Figure 4A). Moreover, the 19 highly abundant vOTUs represented 18 viral genera. NODE_66 and NODE_55b clustered into the same genus-group via vConTACT2. This diverse genus cluster contains as much as 15 vOTUs (species-level groups). Contrastingly, 11 abundant vOTUs are single members of a novel genus group (Figure 4A). They have a genus cluster size of 1, which means that they do not group, at the genus level, with any reference viral sequence. Finally, 4 (21.1%) of the 19 abundant phages were identified as having a lysogenic infection mechanism.

### 3.6. APSE Is the Most Abundant vOTU on the Phyllosphere and Carries a Predicted Insecticidal Toxin

While the majority of the top-rank abundant vOTUs was detected only in either one of the two leaf types, the vOTU classified as Hamiltonella virus APSE-3 was present in both samples. This vOTU had a relative abundance of 23.1% and 3.1% in the FL and PL, respectively (Figure 4A). Different subtypes of APSE phages have been found integrated into *H. defensa* genomes. They display highly conserved genomes, with most variations occurring in two genomic regions used to classify APSE genomes in subtypes, namely a toxin cassette region and the vicinity of the DNA polymerase [48]. The abundant APSE type identified on both FL and PL is predicted to carry a YD-repeat toxin (Figure 3B and Appendix A), placing it in the APSE-3 subtype. The toxin cassette of the wheat APSE has a low amino acid identity compared to other members of the APSE-3 variant. This, along with the proteins present in the DNA polymerase region, suggest the APSE vOTU represents a new subtype: APSE 3.2a (Figure 4B and Appendix A). Interestingly, when mapping the FL and PL reads to the APSE vOTU, there is a notable decrease in coverage at the toxin-cassette region and the vicinity of the DNA polymerase (Appendix A). This suggests that multiple subtypes of the APSE virus might be present on the wheat phyllosphere, including some with additional toxin cassette arrangements.

## 4. Discussion

Viral metagenomes from leaf tissues are challenging to generate due to low microbial density and because the proportion of nucleic acids of viral origin is significantly lower than that of the plant and the microbial fraction [28]. Nevertheless, our virome extraction method was able to recover the viral fraction of the leaf samples quite specifically. This is supported by the high fraction of reads (84.2%) mapping to the predicted viral contigs. 

A high initial amount of plant material was required to obtain sufficient DNA for deep sequencing. This posed two main limitations to this study. Firstly, it had an impact on the resolution, as the samples represent the average community composition of 500 leaves. Secondly, the required scaling-up of experimental procedures was contingent on the available resources, e.g., processing time, lab equipment, etc. This limited the collection of replicates that was required to validate if the viral community differs between leaf types or if the observed differences were due to systematic variance. 

DNA amplification methods, such as MDA, can overcome the input limitations required for sequencing. However, MDA selectively amplifies ssDNA and circular elements [57], and thus, the diversity metrics obtained from an amplified sample might not represent those of the actual viral community. Yet, these methods can reveal different aspects of the viral diversity [50], consistent with our results, where we identified 46 vOTUs unique to the MDA-amplified samples. 

Additionally, one of the main challenges for characterizing viromes is the current limited knowledge of viral genomics [58]. Many bioinformatic analyses, such as viral identification, genome completeness estimation, and taxonomic assignment, depend on a priori knowledge of viral signatures [32,51]. Consistently, as much as 96.8% of the wheat phyllosphere vOTUs represent novel viral species. 

The harsh abiotic conditions of the phyllosphere, e.g., high-UV index, are known to be detrimental to phages [59]. These conditions could therefore select for phages adapted to such pressures. We hypothesized that temperate phages could be enriched due to the protection provided by integrating their DNA into a bacterial host genome. However, most of the recovered high-quality vOTUs (76.2%) were predicted to have a lytic lifestyle. This increased ratio of lytic to lysogenic phages could be explained by the bacterial biomass since low microbial densities seem to favor ‘kill-the-winner’ dynamics, resulting in an increased proportion of lytic phages [60]. 

The presence of an APSE virus amongst the dominant vOTUs of both sample types, suggests they might be ubiquitous members of the wheat virome. However, this phyllosphere virome only represents a snapshot of the viral community. Thus, further studies are required to explore the composition and stability of bacteriophage communities over time. 

*Hamiltonella defensa* is a facultative endosymbiont that colonizes the gut of several species of aphids and whiteflies [61]. It has been described as a member of a tripartite insect–bacteria–phage symbiont system [62]. In this association, the lysogenic APSE virus integrates into the *H. defensa* genome. Then phage-encoded toxins provide the *H. defensa*-colonized aphids with protection from parasitic wasps [48]. APSE viruses can encode one of three types of toxins, including Shiga-like toxin, YD-repeat (RHS-repeat) toxin, or cytolethal distending toxin (CdtB). Notably, APSE, the most abundant vOTU identified on both FL and PL, is predicted to carry YD-repeat toxin (Figure 3B and Appendix A), which is known to provide aphids with the strongest defensive phenotype against parasitic wasps [48]. 

Finally, most of the vOTUs were predicted to interact with the dominant members of the bacterial community. Phylloviruses, as drivers of the microbial community structure [13], could be used for bacteriophage-mediated engineering of the phyllosphere microbiome. Possible applications include either the targeted removal of phytopathogenic bacteria [63,64], or the modulation of the microbial community to increase plant-beneficial bacteria [7]. Therefore, the use of viral metagenomics on phyllosphere research can open a new chapter of plant biology, where bacteriophages are recognized as key members of these microbial communities.

## Figures and Tables

**Figure 1 viruses-14-00244-f001:**
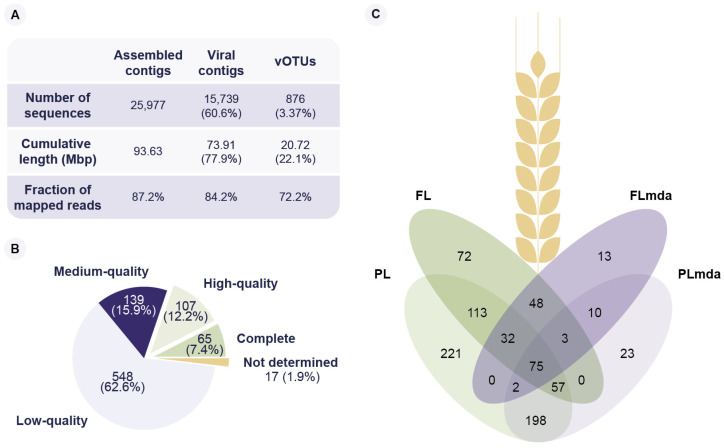
Virus operational taxonomic units (vOTUs) recovered from the four viral metagenomes from winter wheat leaves. Viral metagenomes from flag and penultimate leaves were sequenced without (FL, PL, respectively) and with MDA amplification (FL_mda_, PL_mda_, respectively). (**A**) Overview of total assembled contigs, identified viral contigs, and species-rank viral clusters (vOTUs), in terms of the number of contigs, cumulative contig length, and the fraction of mapped reads. (**B**) Predicted genome completeness of the recovered vOTUs. (**C**) Venn diagram of the presence of vOTUs in the four viral metagenomic samples.

**Figure 2 viruses-14-00244-f002:**
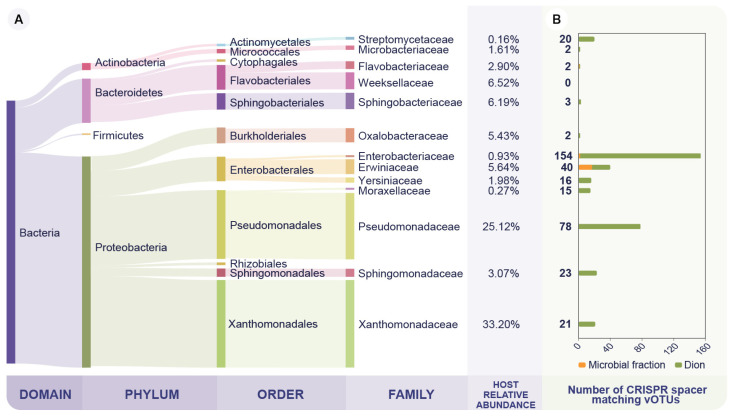
Microbial taxonomic profile and CRISPR-based vOTU host taxonomy assignment. (**A**) Relative abundance of the dominant bacterial families on the bacterial fraction. Taxonomy was assessed at the read level from the microbial fraction samples (FL_mf_ and PL_mf_). (**B**) Host prediction performed by matching the vOTUs against CRISPR spacers either from the Dion CRISPR-spacer database or from the assembled microbial contigs. The y-axis represents bacterial families and the x-axis the number of vOTUs predicted to infect such family. Notably, Moraxellaceae and Streptomicetaceae are not highly abundant families in the microbial fraction. However, they were included in the figure, since ≥15 vOTUs were predicted to infect these bacterial families.

**Figure 3 viruses-14-00244-f003:**
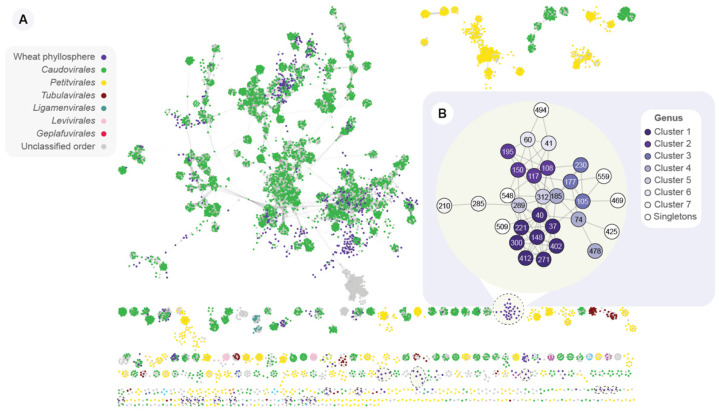
Protein sharing network of wheat phyllosphere vOTUs, viral RefSeq genomes, and IMG/VR viral sequences. (**A**) Overview of the vConTACT2 protein-profile network, depicting the taxonomic context of phylloviruses. In this network, nodes represent viral sequences, and edges connecting two nodes indicate a shared protein profile. RefSeq nodes are colored according to their taxonomy at the order level. Dashed-line circles indicate the 17 discrete groups of interconnected nodes containing only wheat phyllosphere vOTUs. (**B**) Zoom-in on the biggest group of interconnected nodes containing only wheat phyllosphere. Nodes are colored according to the assigned vConTACT2 viral cluster, which delineates clusters at approximately the genus level.

**Figure 4 viruses-14-00244-f004:**
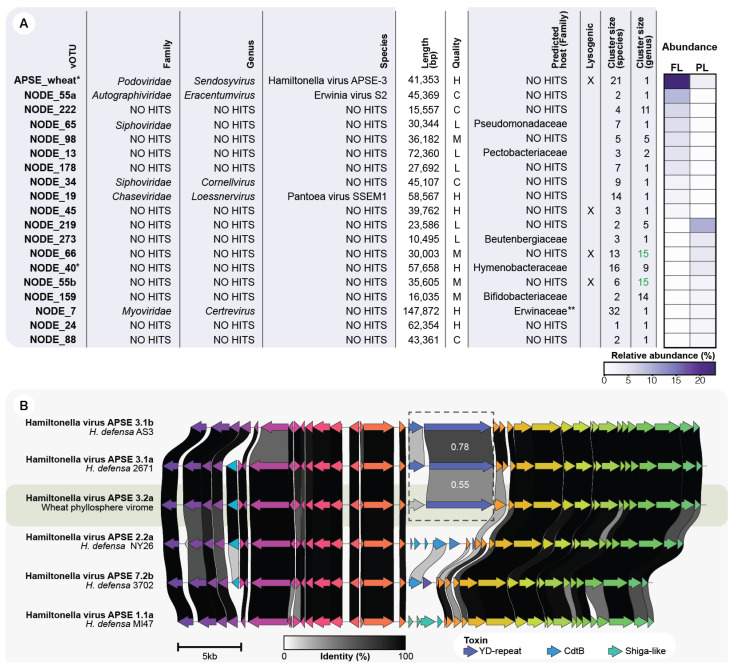
(**A**) Description of the ten top-rank abundant phages of each leaf sample (FL and PL). Taxonomic assignments at genus and family levels were made using vConTACT2 and at species level using nucleotide clustering at >95% identity and >85% sequence coverage. The species cluster size represents the number of viral contigs clustered in a vOTU (species-level). On the other hand, genus cluster size refers to the number of viral species (vOTUs) belonging to the same genus cluster assigned by vConTACT2: (C) Complete, (H) High quality, (M) Medium quality, (L) Low-quality, and (U) Unclassified. * Contigs assembled from a subsampled dataset, ** Viral contig matching a CRISPR spacer predicted from the microbial fraction assembly. Green font indicates the two vOTUs that are part of the same genus cluster. (**B**) Comparative genomics between the wheat phyllosphere APSE phage and the reference genomes of the APSE-1, APSE-2, APSE-3, and APSE-7 subtypes. The amino acid identity between the YD-repeat toxins is indicated in white numbers, and predicted open reading frames are colored by protein groups.

## Data Availability

The sequencing data used and described in this study was submitted to NCBI under the BioProject accession number PRJNA733924.

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
