# Peer review of "Bacteriophages Roam the Wheat Phyllosphere"

_viruses, 2022, doi:10.3390/v14020244_

Round 1

Reviewer 1 Report

Please see attached documents for comments to authors

Reviewer 2 Report

This study presents the analysis of the wheat phyllosphere’s virome linked to the bacterial community present in the same niche. A better understanding of microbial interactions in the environment is very important; however, this manuscript requires some attention particularly in the methods sections to confirm the robustness of the analysis presented. Find some specific questions and suggestions below.

The term phyllosphere needs to be defined in the manuscript text (abstract is separate).

The introduction needs to be a little more informative to establish the importance of the work presented:

- Lines 32-34. “colonized by many epiphytes, even though...” - the logical nexus here is unclear.

- Why is it important to define the virome of the phyllosphere? How do phages impact bacterial populations? What are the “unanswered questions” (line 44)?

- Why had no phyllosphere viral metagenome been sequenced (lines 39-40) before? What are the challenges for this type of study, and consequently what are the biases or weaknesses of the present study? these should be clearly outlined in Introduction (the former) and Discussion (the latter).

- Lines 53-54. Unclear sentence. To what does “remaining fraction” refer?

The methodology requires more detail:

- Line 58. (ref) insert reference?

- How does the chosen amplification protocol (Cytiva illustraTM Ready-To-GoTM GenomiPhi V3 DNA Amplification Kit Virome) bias metagenome recovery? Has this approach been used before? Has it been validated?

- Line 78. Are these the same pellets as in line 64? If yes, should this be ‘microbial pellets’?

- Was the microbial DNA sequenced by shotgun sequencing? If so, the analysis protocol needs to be outlined. What does “WGS” (line 83) refer to?

- Lines 86-87. What do you mean by “amplified and non -amplified viromes”?

- Multiple displacement amplification (MDA) is not described.

- Line 97. How was the target coverage calculated?

- Line 100. “Contigs assembled using 100% of the reads” - this is unclear.

- What is the clustering into different vOTUs, which are then separately classified as linked to different species or viral types, based on? Lines 107-108. Is the clustering based on a reference database? If so, which one? if not, how are the reads binned?

- Lines 104-114. There is some confusion here with the use of the terms ‘contigs’ ‘sequences’ and ‘genomes’ which are not interchangeable. Could this be verified and reworded were necessary?

- Lines 125-138. This should go with the above section. It seems that there is repeated information re. vOTU picking and clustering in this paragraph and the above.

- Lines 119-120. Unclear. What is the genome of a contig?

- Line 153. “average length” - this seems rather arbitrary as this depends on the phage type/family and ranges in most cases between 40 and 200 kb. Is the 50 kbp figure meant to be just an example?

Results:

- How was the microbial community profiled?

- How can the samples be compared when they have not been normalized (input DNA amount very different - Table S1)?

Round 2

Reviewer 2 Report

The authors made extensive changes to the manuscript that have improved its readability and have provided satisfactory answers to most original queries. There is one question left to address and some minor corrections/edits that need attention. Find my comment in the marked-up pdf.
